# The Onset of Dental Erosion Caused by Food and Drinks and the Preventive Effect of Alkaline Ionized Water

**DOI:** 10.3390/nu13103440

**Published:** 2021-09-28

**Authors:** Tsutomu Sato, Yoshitaka Fukuzawa, Satoshi Kawakami, Megumi Suzuki, Yoshinori Tanaka, Hayato Terayama, Kou Sakabe

**Affiliations:** 1Division of Basic Medical Science, Tokai University School of Medicine, Isehara 259-1193, Japan; kawakamis@tsc.u-tokai.ac.jp (S.K.); ksato21@aol.com (Y.T.); terahaya@tokai-u.jp (H.T.); sakabek@tokai-u.jp (K.S.); 2Division of Basic Research, Functional Water Research Laboratory, Louis Pasteur Center for Medical Research, Kyoto 606-8225, Japan; 3Aichi Medical Preemptive and Integrative Medicine Center, Aichi Medical University Hospital, Nagakute 480-1103, Japan; yofuku@aichi-med-u.ac.jp; 4Department of Dental Hygiene, The Nippon Dental University College at Tokyo, Tokyo 102-0071, Japan; megumi-t@tandai.ndu.ac.jp

**Keywords:** acidic foods and drinks, nutrients, dental erosion, alkaline ionized water, demineralization, oral flora

## Abstract

In recent years, the incidence of dental erosion caused by the ingestion of acidic foods and drinks, including sports drinks, has been increasing in Japan and elsewhere. Therefore, the problems associated with this injury can no longer be ignored in dental clinical practice. The ingestion of these foods and drinks is important from the viewpoint of overall health and disease prevention. For example, fermented foods, such as Japanese pickles, enhance the nutritional value of foodstuffs and promote the absorption of nutrients into the body, and sports drinks are useful for preventing heat stroke and dehydration. Therefore, eliminating these intakes is not a viable solution. In this paper, we outline the mechanism of dental erosion caused by acidic beverages and also describe the effectiveness of alkaline ionized water (AIW) at preventing acid erosion. Given the fact that the complete elimination of acidic beverage consumption is highly unlikely, remedies such as the use of alkaline ionized water (AIW) may be helpful.

## 1. Dental Caries and Acid Erosion

Dental caries is an infectious disease caused by cariogenic bacteria that inhabit the oral cavity, and its pathology involves the demineralization of dental hard tissues by the acids produced by cariogenic bacteria [1]. Furthermore, since the occurrence of dental caries is related to lifestyle-related factors, such as diet and tooth brushing habits, it can also be characterized as a lifestyle-related disease. The prevalence of dental caries in Japan is still high compared to other industrialized countries, but it is gradually decreasing. Similar to dental caries, dental acid erosion is a disease that causes demineralization of the teeth. Acid erosion is a disease in which the enamel formed on the tooth surface undergoes demineralization due to organic substances present in the environment, especially acids, and has been known as an occupational disease for a long time [2]. In other words, there are no bacterial acids involved in the development of dental acid erosion. Dental erosion is found in workers in manufacturing industries that have long used or produced acids, such as sulfuric acid, nitric acid, and hydrochloric acid. It is also seen in workers who are employed in factories that produce dyes and gunpowder, metal plating, storage batteries, and fertilizers. Teeth that have been decalcified by chemical acids are called erosive teeth. On the other hand, in recent years, apart from occupational dental erosion, epidemiological surveys in Japan and overseas have revealed that the incidence of dental erosion caused by the intake of highly acidic foods and drinks has been increasing. Recently, Japan has been reporting higher incidences of dental erosion among athletes, infants, and the elderly due to their frequent consumption of sports drinks for dehydration purposes and to prevent heat stroke [3]. We measured the pH of several types of beverages that are often consumed in Japan [4] and observed that most were below pH 5.6–5.7, which is the pH at which enamel is decalcified [5,6,7,8] (Table 1).

The recent increased consumption of acidic foods and drinks is a major health concern. For example, fermented foods enhance the nutritional value of foodstuffs and promote the absorption of nutrients, and sports drinks are useful for preventing heat stroke. Therefore, it is not appropriate to avoid these intakes. In this paper, we describe the dental erosion caused by acidic beverages and discuss the effectiveness of preventing erosion with alkaline ionized water (AIW).

## 2. Alkaline Ionized Water

Alkaline ionized water (AIW) is generated by electrolyzing tap water at the cathode of a household electrolyzed water generator (alkaline ionized water conditioner) that has JIS (Japanese Industrial Standards; JIS T2004) approval [9]. It is weakly alkaline (pH 9–10) water. AIW has been confirmed to be effective at improving gastrointestinal symptoms [10] and has been approved for drinking by the Japanese Ministry of Health, Labor and Welfare. We have found that AIW is also useful in maintaining oral health [11], and in this paper, we describe the usefulness of AIW (pH 9.5) in the prevention of dental erosion of teeth based on the results of our previous research.

## 3. Measuring Enamel Surface pH

The measurement of saliva pH is a useful test for assessing the risk of enamel demineralization and is widely used [12]. However, due to the buffer capacity of saliva [13,14,15], it is possible that its pH does not always reflect the pH of the enamel surface (Es pH). Therefore, we have established a system for measuring the Es pH [16]. The Es pH was measured using a continuous measurement system consisting of an oral pH antimony electrode (model: SP-Sb-052, Chemical Equipment Co., Ltd. Tokyo, Japan), a pH meter (model: PH-201Z, Chemical Equipment Co., Ltd.), a recorder (model: VR-71, T & D Co., Ltd. Nagano, Japan), a USB serial converter (Latoc System Co., Ltd. Osaka, Japan), and a personal computer (Figure 1). The measurement site of the Es pH was the buccal side of the left molar of the maxilla, and the measurements were performed at rest and before and after ingesting an acidic beverage and AIW. The Es pH was measured once in each experiment. When ingesting a beverage, the electrode is not initially in contact with the enamel surface. The electrode is brought into contact with the tooth surface quickly after intake, which takes about 10 s. Figure 2, Figure 3, Figure 4, Figure 5 and Figure 6 show the measurement results of the Es pH, and the values are presented as continuous data for convenience. The experiment was conducted 3 h after brushing following lunch. As a result of measuring the Es pH of resting subjects (N = 5) for 5 min continuously, the pH did not always show a constant value but showed a change in the range of about 0.1 to 0.3. Therefore, in this experiment, measurements were taken continuously for 5 min, and the median of the minimum and maximum values observed during this period was taken as the Es pH at rest. As a result, individual differences were observed in the Es pH, and the median was in the range of 5.2–5.9 (Table 2). Figure 2 shows the measurement results of the Es pH of subject ID 5 at rest. At the measurement time of 5 min, the pH fluctuated between approximately 5.7 and 6.1.

## 4. Es pH after Ingesting Acid Beverages and AIW

The acidic beverages used in the experiments were cola and a sports drink, with pH values of 2.2 and 3.3, respectively. The method of ingestion of the acidic beverage involved the subject keeping 50 mL of it in their mouth while the pH electrode was attached to the surface of the enamel and then swallowing the beverage after swishing it around the oral cavity. For the ingestion of the AIW, when the Es pH reached the lowest value after ingesting the acidic beverage, the pH electrode was removed from the tooth surface, and 50 mL of the AIW was immediately ingested in the same manner as for the acidic beverage. After ingesting the AIW, the pH electrode was placed on the tooth surface again, and the pH was continuously measured. Measurements were also made using tap water (pH 6.7) instead of AIW.

After ingesting cola, the Es pH of 5 subjects decreased from 3.1 to 3.3 (Table 3) and then gradually increased. Although the rate of pH increase was different among the subjects, it was confirmed that the plateau was almost reached in all the subjects after about 6–8 min. The pH at the time of reaching the plateau was close to 7 in all subjects, which was higher than the pH before the ingestion of cola. This phenomenon was considered to be due to the buffer capacity of saliva released after being stimulated by ingesting beverages [13,14,15]. When the increase in the Es pH reached a plateau, the change in the Es pH was measured when cola was ingested again and then the AIW was ingested. The Es pH decreased after the ingestion of the acidic beverages but increased rapidly with the subsequent ingestion of the AIW (Figure 3). Then, in all the subjects, the pH was close to 7 after about 10–20 s. In an experiment using tap water instead of AIW, the re-elevation of the Es pH was slower than that of the AIW, and it took about 9–12 min to reach approximately pH 7 (Figure 4). The Es pH after ingesting the sports drink decreased from 3.5 to 3.9 (Table 3) and then gradually increased. Similar to cola, the rate of the pH rise was different in the subjects but almost reached a plateau after about 10–12 min. The pH at the time of reaching the plateau was close to 7 in all subjects, which was higher than the pH before the ingestion of the sports drink. When the Es pH reached the plateau, the change in the Es pH was measured when the sports drink was ingested and then the AIW was ingested. As a result, as in the case of cola, the Es pH decreased after the ingestion of the sports drink but increased rapidly with the subsequent ingestion of the AIW (Figure 5). All subjects exhibited a pH close to 7 after about 15–20 s, following which a plateau was reached. In the experiment using tap water instead of AIW, the re-increase in the Es pH was gradual, similar to cola, and it took about 12–15 min to reach approximately pH 7 (Figure 6). Since almost the same pattern was observed in all the subjects with respect to the change in the Es pH when ingesting the acidic beverage or AIW shown above, only the data for subject ID 4 are presented.

## 5. Discussion

The pH of the acidic beverages used in our experiment was 2.2 for cola and 3.3 for the sports drink. Since the critical pH of enamel decalcification is considered to be approximately 5.5–5.7 [5,6,7,8], acidic beverages will increase the likelihood of dental erosion or enamel demineralization/decalcification. In recent years, hydration with sports drinks has been widely recommended to prevent heat stroke and dehydration, and people of all ages consume sports drinks in Japan. Therefore, it is believed that the risk of dental erosion will increase in the future. It is important to note the frequency and number of intakes of acidic beverages in order to prevent dental erosion. In particular, since it is considered that the decrease in the Es pH directly indicates the risk of developing dental erosion, it is necessary to promptly increase the Es pH that is decreased by the ingestion of acidic beverages. In our study, the degree of the decrease in the Es pH after ingestion and the rate of the subsequent increase were different between cola and the sports drink. That is, the Es pH was lower in cola at pH 2.2 than in the sports drink with pH 3.3. Based on this finding, we believe that the degree of the decrease in the Es pH after the ingestion of the acidic beverage was related to the pH of the beverage. The Es pH, which decreased with the ingestion of cola or the sports drink, subsequently increased and reached a plateau at a value close to 7 in all subjects. The pH value of 7 is higher than the pH before the ingestion of the acidic drink, and we believe this was because the stimulation by the acidic drink temporarily promoted saliva secretion and the increased saliva came into contact with the enamel. We observed this when the Es pH decreased due to the ingestion of the acidic beverages; the Es pH increased rapidly following AIW ingestion. This increase was also observed when tap water was ingested but at a slower rate than that of the AIW. Therefore, when ingesting an acidic beverage, it is believed that the continuous ingestion of AIW afterward is effective in preventing dental erosion. It was also observed that the rate of increase in the Es pH by the AIW was slower for the sports drink than for cola. Although the cause is unclear, it has been reported that the pattern of changes in the Es pH due to acidic beverages differs depending on the type of beverage [4]. Moreover, this difference seems to be important in assessing the risk of an erosive effect on teeth. That is, the risk of dental erosion may be higher with sports drinks than cola. In the future, we plan to add other acidic beverages and further investigate the changes in the Es pH and the effects of AIW on them when they are ingested.

In this paper, we described the preventive effect of AIW on dental erosion; however, it is also known that AIW is effective at improving oral flora [18,19] and intestinal flora [20,21]. Therefore, AIW may contribute to reducing the risk of diseases associated with these flora.

## 6. Conclusions

We investigated the effectiveness of alkaline ionized water (AIW) in preventing the occurrence of dental erosion caused by acidic beverages. We confirmed that the Es pH, which was lowered by the ingestion of cola or a sports drink, was rapidly increased by the ingestion of AIW. Therefore, AIW was shown to be useful in preventing dental erosion caused by acidic beverages. We recommend the intake of a sufficient amount of AIW after ingesting acidic foods and drinks.

## Figures and Tables

**Figure 1 nutrients-13-03440-f001:**
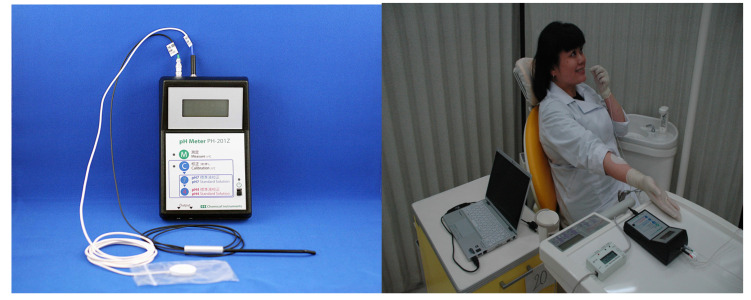
Measurement of enamel surface pH [13]. **Left**: pH antimony electrode connected to pH meter. **Right**: Photograph taken during enamel surface pH measurement.

**Figure 2 nutrients-13-03440-f002:**
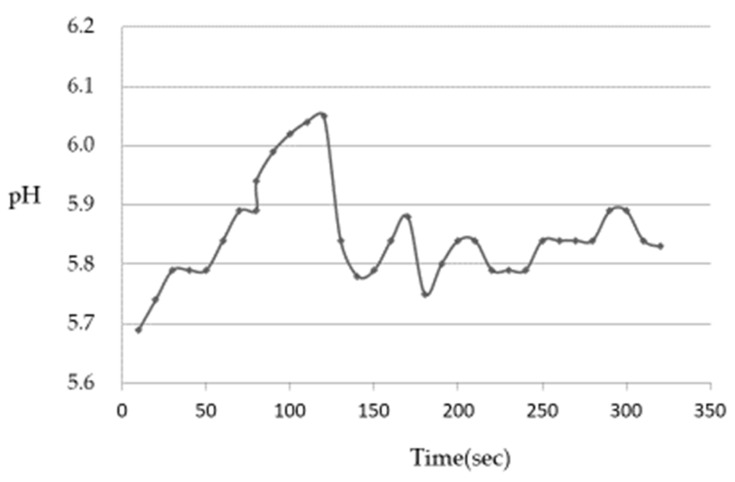
Changes in Es pH at rest (Subject ID:5) [17].

**Figure 3 nutrients-13-03440-f003:**
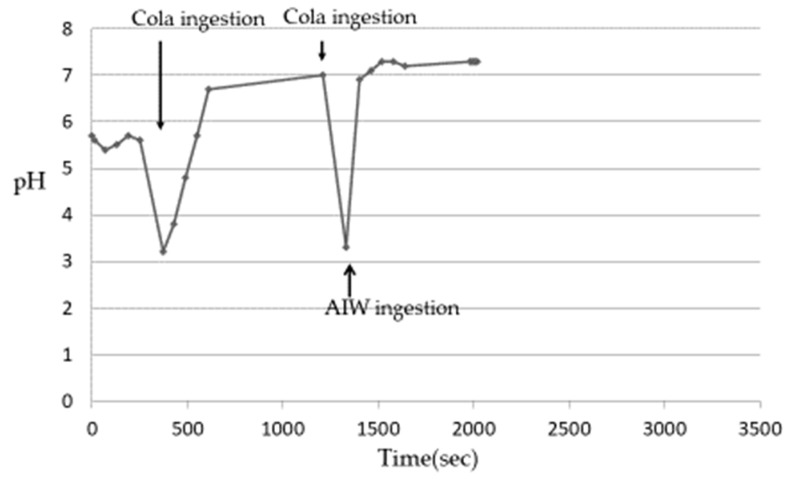
Changes in Es pH after ingestion of cola or AIW (alkaline ionized water) (Subject ID:4) [17].

**Figure 4 nutrients-13-03440-f004:**
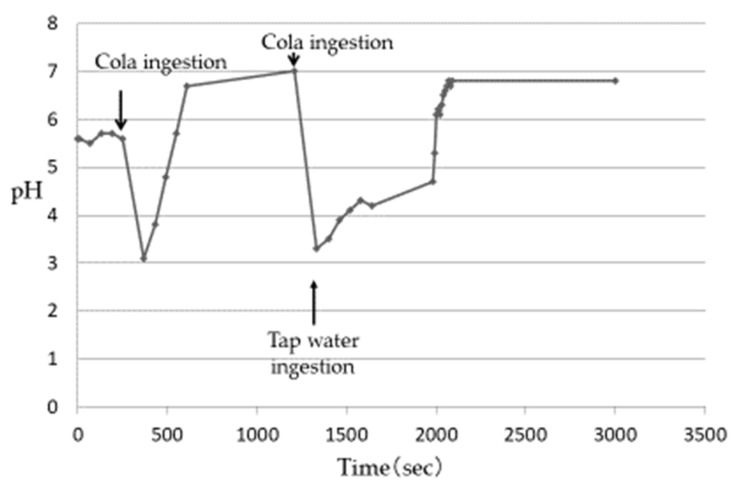
Changes in Es pH after ingestion of cola or Tap water (Subject ID:4) [17].

**Figure 5 nutrients-13-03440-f005:**
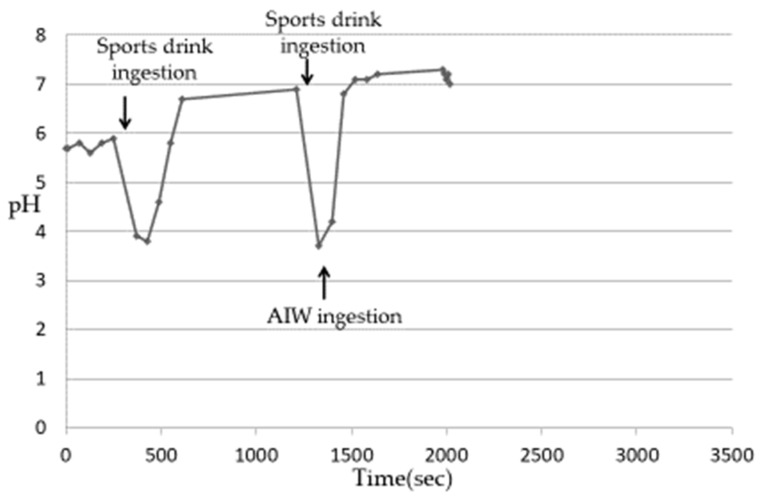
Changes in Es pH after ingestion of sports drink or AIW (Subject ID:4) [17].

**Figure 6 nutrients-13-03440-f006:**
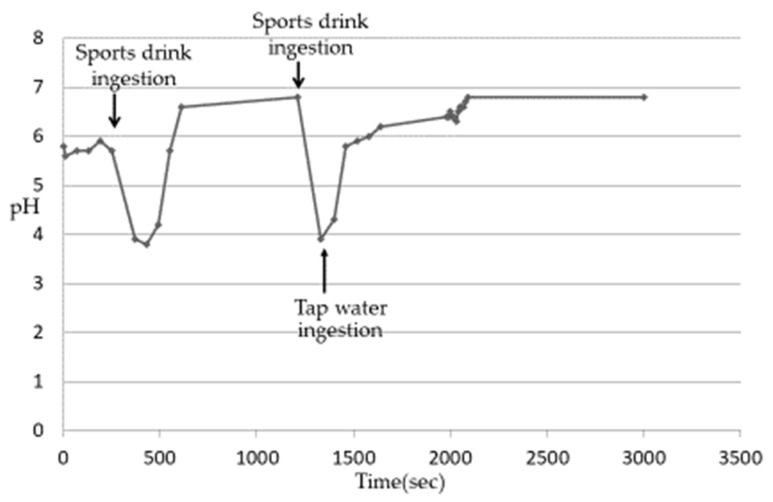
Changes in Es pH after ingestion of sports drink or Tap water (Subject ID:4) [17].

**Table 1 nutrients-13-03440-t001:** pH values of various beverages [4].

Beverage	pH
Cola	2.2
Sports drink	3.3
Orange juice	3.7
Yoghurt drink	3.9
Barley tea (Japanese mugi cha)	6.1

**Table 2 nutrients-13-03440-t002:** Es pH (the pH of the enamel surface) with the subjects at rest [17].

Subject ID	pH (Median)
1	5.5
2	5.2
3	5.8
4	5.7
5	5.9

**Table 3 nutrients-13-03440-t003:** Decrease in Es pH after ingestion of beverage [17].

Subject ID	Cola	Sports Drink
1	3.1	3.9
2	3.1	3.6
3	3.2	3.5
4	3.1	3.6
5	3.3	3.8

The numbers represent the lowest value.

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
