# Peer review of "The Onset of Dental Erosion Caused by Food and Drinks and the Preventive Effect of Alkaline Ionized Water"

_nutrients, 2021, doi:10.3390/nu13103440_

Round 1

Reviewer 1 Report

Regarding the entire manuscript, please check spaces and fonts and more importantly grammar.

The term erosive teeth can also be stated as dental erosion- you mention erosive teeth through the entire manuscript.

Regarding Methodology: It is not clear how many experiments were conducted. The few experiments included should have a section on limitation of the study. The connection to lowering or eliminating cancers is very weak. That association between AIW use and lowered cancer rates either needs further explanation or a deliberate elimination.

The manuscript explains the rationale for using Alkaline Ionized Water, the experiments and diagrams are clear. However, the explanation and cause/effect strategy are not fully revealed in the current version. In other words, the manuscript needs to be fully edited by a native English speaker if it is going to published in English.

Abstract: 

Regarding the sentence, Ingestion of these foods and drinks is important from the viewpoint of nutrition and disease prevention.

Perhaps you can include health as in: Ingestion of these foods and drinks are important from the viewpoint of overall health and disease prevention.

This sentence needs to be restructured; Therefore, it is not appropriate to deny these intakes. In this paper, we outline the mechanism of acid erosion caused by acidic beverages and also explain the usefulness in prevention of acid erosion with alkaline ionized water. As a result, it was suggested that the oral flora was improved, leading to the prevention of lung cancer.

Perhaps you meant: Given the fact that the elimination of the acidic beverages is highly unlikely, remedies such as the use of alkaline ionized water are helpful. Alkaline ionized water has known to improve oral flora that leads to lung cancer prevention.

Under 1. Dental caries and acid erosion; should this be 1. Dental Caries and Acid Erosion (capital letters?)

Page 1-Lines 34-36 the statement of "The acid erosion  is a disease in which the enamel formed on the tooth surface causes demineralization due  to the organic substances existing in the working environment, especially acid, and has been known as an occupational disease for a long time〔2〕." Please restructure this statement.

Line 37 regarding:

That is, there is no bacterial  acids in the development of the acid erosion: this statement does not make sense, please restructure it.

Lines 38-41

Typical occupations that develop the acid erosion are found in industries that have processes such as manufacturing acids such as sulfuric acid, nitric acid, and hydrochloric acid, manufacturing dyes and gunpowder,  plating, storage batteries, and fertilizer factories. The tooth that have been decalcified by chemical acids are called erosive teeth.

You can shorten the first sentence by beginning with Manufacturing industries have long produced....

The second sentence does not make sense, please reword it.

Lines 42-43

You can eliminate this sentence that is not connecting to any concept

In developed countries, the incidence of acid erosion is decreasing due to the establishment of occupational health control.

Page 2 Lines 46-48

In Japan as well, there is concern about the occurrence of acid erosion teeth, especially among athletes, infants and the elderly who often consume sports drinks for the purpose of preventing heat stroke and dehydration, and countermeasures are an urgent issue〔3〕

You may re word it as: Recently Japan is reporting higher incidences of  dental erosion amongst athletes, infants and elderly due to frequent consumptions of sport drinks for stroke and dehydration purposes.

Lines 52-53

As mentioned above, intake of these foods and drinks is important from the viewpoint of nutrition and disease prevention.

Do you mean: The recent increase consumption of acidic foods and drinks is of a major health concern.

Table 1 regarding Table 1. pH of beverages 〔4〕.

The reviewer believes that you meant Beverage and not Beberage and for Cola you can also mention carbonated beverages

Under Discussion

Page 7 Lines 153-154; ...these beverages are considered to be at risk of enamel decalcification.

Did you mean Acidic beverages will increase the likelihood of dental erosion or enamel demineralization/decalcification?

Lines 156-158

In order to prevent acid erosion, it is important to pay attention to the number of intakes and the frequency of  intake of highly acidic beverages and to shorten the time when saliva and Es pH are low  as much as possible.

You can state it as:

It is important to note the frequency and duration of acidic beverages in order to prevent dental erosion.

Page 8 Lines 182-185 regarding

and intestinal flora〔20,21 182 〕. Therefore, AEW may contribute to reducing the risk of diseases associated with these 183 flora. For example, periodontal disease caused by oral flora has been reported to increase 184 the risk of colorectal〔22,23〕and lung cancer〔23〕. In the future, we would like to examine 185 the cancer prevention effect of daily intake of AEW.

This manuscript did not fully explain the association between periodontal disease, oral flora and increase risk of lung and colorectal cancers with the exception of this line under discussion. This association should either be removed or elaborated further.

Perhaps a diagram could explain what happens with the consumption of carbonated beverages, the pH effect and end results.

Good luck and thank you for allowing this reviewer to read your manuscript.

Author Response

I  reply to reviewer's comments as follows.

  1. The revised manuscript was checked by a native English speaker.
  2. changed the title of MS from "The Onset of Acid Erosion Caused・・・"to "The  Onset of Dental Acid Erosion Caused・・・.
  3. Regarding Methodology: It was added that the Es pH was measured once in each experiment.
  4. Abstract: The sentence that was pointed out was restructured.
  5. Removed the statement of cancer and also removed the discussion of the link to cancer.
  6. Page 1-Lines 34-36  Restructured the statement.
  7. Line 37 Restructured the sentence.
  8. Lines 38-41 Restructured and shortened the sentences.
  9. Lines 42-43 Eliminated the sentence.
  10. Page 2 Lines 46-48 Re worded the sentences point outed.
  11. Line 52-53 Corrected the sentence.
  12. Table 1 Corrected spelling  (Beverage)
  13. Discussion Page Lines 153-154 Restructured the sentence.
  14. Lines 156-158 Restructured the statement.
  15. Page 8 Lines 182-185 Removed the sentence link to cancer.

Reviewer 2 Report

The paper "The Onset of Acid Erosion Caused by Food and Drink and the Preventive Effect of Alkaline Ionized Water" is an interesting paper for investigating the role of alkaline water in neutralizing acids and preventing acid erosion and cavity. It is important to underline in the introduction the role of alkaline ionized water in neutralizing the acidity associated with cariogenic bacteria. Alkaline ionized water can neutralize some of this acidity while helping remineralize your teeth due to the additional calcium it contains.

Author Response

In this paper, we stated that AIW is effective in preventing dental acid erosion caused by acidic beverages. I would like to discuss the relationship between caries caused by acid of cariogenic bacteria and AIW in another paper.

Round 2

Reviewer 1 Report

Thank you for your updates and modifications. Please remove the word acid from everywhere you mention dental acid erosion and change it to either enamel erosion or dental erosion. Please check spaces in between words.

The paper still needs to be edited by a native English speaker. The formatting is not standard.

Author Response

I completely removed the word acid from the text, and changed it to dental erosion. I checked spaces in between word. The text was  checked by two native English speakers. 

Thank you for your understanding.